# Effect of Ferulic Acid on Semen Quality of Goat Bucks during Liquid Storage at 17 °C

**DOI:** 10.3390/ani13152469

**Published:** 2023-07-31

**Authors:** Feng Zhang, Shichang Han, Nian Zhang, Jin Chai, Qi Xiong

**Affiliations:** 1Hubei Key Laboratory of Animal Embryo Engineering and Molecular Breeding, Institute of Animal Husbandry and Veterinary, Hubei Academy of Agricultural Sciences, Wuhan 430064, China; zhangfeng0130@163.com (F.Z.); hanshic1235@163.com (S.H.); zhangnian@hbaas.com (N.Z.); 2Key Laboratory of Swine Genetics and Breeding of the Agricultural Ministry, Key Laboratory of Agricultural Animal Genetics, Breeding and Reproduction of the Ministry of Education, College of Animal Science and Technology, Huazhong Agricultural University, Wuhan 430070, China

**Keywords:** ferulic acid, goat, liquid storage, 17 °C, semen quality

## Abstract

**Simple Summary:**

This study aims to explore whether ferulic acid (FA), as an exogenous antioxidant, could improve the quality of goat semen during liquid storage at 17 °C. Semen was collected from three black-headed goat bucks and stored at 17 °C with different concentrations of FA. The effect of FA on semen quality was evaluated via semen quality detection and antioxidant index analysis. The results showed that adding 50 μmol/L of FA significantly improved the semen quality from 1 to 5 days. Furthermore, supplementing semen with 50 μmol/L of FA preserved at 17 °C for 3 days had no significant effect on fertility. Overall, adding 50 μmol/L of FA in dilution improved the quality of goat semen stored at 17 °C.

**Abstract:**

This study investigated the effect of different concentrations of ferulic acid (FA) on the quality of goat semen preserved at 17 °C. First, semen was collected from three black-headed goat bucks using an artificial vagina. Then, the mixed semen was diluted with basal dilutions containing different concentrations of FA (0, 25, 50, 100, and 200 μmol/L) and stored at 17 °C. Sperm total motility, plasma membrane integrity, acrosome integrity, reactive oxygen species (ROS) levels, malondialdehyde (MDA) content, and total antioxidant capacity (T-AOC) were measured during semen storage. The results showed that sperm total motility, plasma membrane integrity, and acrosome integrity were significantly improved in the 50 μmol/L FA group compared with the control group (0 μmol/L) on days 1–5, and the level of T-AOC significantly increased, while the contents of ROS and MDA significantly reduced. Meanwhile, the goats’ conception rate showed that supplementing semen with 50 μmol/L FA preserved at 17 °C for 3 days had no significant effect on fertility. Taken together, our findings suggest that adding 50 μmol/L FA in dilution at 17 °C can improve goat bucks’ semen quality.

## 1. Introduction

With the development of science and technology, artificial insemination (AI) is seeing wider use in goat breeding. The success of AI is largely dependent on the quality of preserved semen, which is a key factor affecting its effectiveness [1,2]. However, compared with other livestock, semen preservation in goats is less studied, and the technology is still not mature. Thus, we are seeking methods that can be applied to the preservation of goat buck semen. Semen preservation technology allows semen to be preserved for longer by maintaining the functional, ultrastructural, and biochemical properties of the sperm after a specific treatment, followed by preservation in the corresponding environment [3]. It can be divided into liquid and frozen preservation, according to the preservation temperature [4]. Liquid preservation has the advantages of simple operation, low requirements for preservation conditions, and a good effect on fertility compared with frozen preservation. Therefore, the investigation of goat buck semen stored in a liquid state is important to goat AI.

Semen quality degradation is inevitable during storage, and one of the main reasons for this is oxidative damage. Reactive oxygen species (ROS) are products of the aerobic metabolism of sperm. Low concentrations of ROS are essential for sperm function, such as sperm capacitation, acrosome reaction, sperm–egg binding, and related signaling pathways [5]. However, high concentrations of ROS reduce sperm viability, plasma membrane integrity, and increase DNA damage through lipid peroxidation, leading to cell apoptosis, and reducing semen preservation quality [6,7]. In general, various enzyme systems involved in redox homeostasis regulate the ROS level of semen, whereas oxidative stress occurs as a result of oxidant excess, antioxidant deficiency, or both [8]. There are antioxidant defense balance systems in semen, including superoxide dismutase, catalase, glutathione peroxidase, glutathione reductase, and nonenzymatic antioxidants such as methionine, ascorbic acid, and α-tocopherol, etc. [9,10,11,12]. When semen is preserved in vitro, the balance is easily disrupted, so it is necessary to add antioxidants or other protective agents to semen to avoid oxidative stress damage [2].

Ferulic acid (FA), a natural phenolic phytochemical widely present in plants, is an important antioxidant. Previous studies have reported that FA plays a positive role in the preservation of rooster [13], boar [14], and Qezel ram [15] semen. However, to our knowledge, the effect of FA on the quality of goat buck semen during liquid storage has not been reported. Therefore, to test whether FA can improve the preservation quality of goat buck semen during liquid storage, FA of different concentrations was added to the semen diluent of goats and stored at 17 °C. Then, the effect of FA on semen quality was evaluated via semen quality detection and antioxidant index analysis. Finally, the optimal concentration of FA for semen preservation at 17 °C was determined.

## 2. Materials and Methods

### 2.1. Semen Collection and Liquid Preservation

The semen of three black-headed goat bucks (3–5 years old) was used in this study. The black-headed goat is a new breed based on Macheng Black Goat (Chinese local breed) and introduces Boer Goat lineage. It has a black head and a white body. Prior to this study, goat bucks were ejaculated two times per week during the breeding seasons. In this study, one ejaculate from each goat buck was collected using an artificial vagina in the presence of estrus goats. The semen volume of each goat buck collected was from 0.5~1.5 mL every time. The semen used was milky white and had no abnormal smell. The sperm concentration was >2 × 10^9^ sperm/mL, and the total motility was >0.8. The semen was pooled to minimize individual differences between goat bucks.

Tris-based solution composed of Tris 250 mmol/L, fructose 100 mmol/L, sodium citrate 75 mmol/L, penicillin 50,000 IU, streptomycin 50,000 IU, and double-distilled water 100 mL (chemicals all from Sigma, St. Louis, MO, USA) was used as the base extender. The mixed semen was diluted 10-fold with diluent and divided equally into five aliquots. FA was added to the base extender at concentrations of 25, 50, 100, and 200 μmol/L, while the control was the base extender without FA. All samples were stored in a constant temperature refrigerator at 17 °C (shaken and turned over every 12 h) and used for the experiment.

### 2.2. Sperm Total Motility Analysis

Every 24 h, 10 μL of semen was taken, respectively, from four FA treatments and one control group, shaken slowly before use, diluted 10 times by adding isothermal base diluent, and incubated in a 37 °C water bath for 3 min. After incubation, 5 μL of the semen was added dropwise to a prewarmed slide, covered with a coverslip, and placed on a HT CASA II automatic sperm analyzer (Hamilton Thorne, Beverly, MA, USA) thermostatic loading table to assess sperm total motility. Five or more fields of view were randomly selected, and the total number of sperm observed was at least 800. There were three replicates per group. 

### 2.3. Sperm Plasma Membrane Integrity Analysis

Plasma membrane integrity assay was performed with a commercial kit (Solarbio, Beijing, China). In brief, 20 μL of semen was taken, respectively, from four FA treatments and one control group and added to 200 μL of hypotonic solution. The solution was prewarmed, stirred gently, and incubated for 30 min at 37 °C. After incubation, 10 μL of the mixture was spread evenly on a slide for observation via light microscopy. Three or more fields of view were randomly selected, and the total number of sperm observed was at least 200. There were three replicates per group. 

### 2.4. Sperm Acrosome Integrity Analysis

Semen from four FA treatments and one control group was used to analyze the sperm acrosome integrity. Fluorescein isothiocyanate-peanut agglutinin (FITC-PNA, Sigma, St. Louis, MO, USA) combined with 4,6-diamino-2-phenyl indole (DAPI, Beyotime (Shanghai, China)) was used to detect the sperm acrosome integrity. Additionally, 10 μL of semen was resuspended with phosphate-buffered saline (PBS) to a concentration of 1 × 10^6^ sperm/mL, and then 10 μL of FITC-PNA staining solution was added and incubated in darkness for 10 min at 37 °C. After incubation, sperm cells were collected via centrifugation at 1500× *g* for 5 min; this process was repeated 1–2 times to wash off the floating color. The sperm cells were then fixed in 4% paraformaldehyde at room temperature for 30 min. After fixation, the cells were washed with PBS and incubated with DAPI. FITC-PNA staining was judged by the criteria that sperm with broken acrosomes were stained with green fluorescence, while sperm without green fluorescence had intact acrosomes. Three or more fields of view were randomly selected, and the total number of sperm observed was at least 200. There were three replicates per group. 

### 2.5. ROS, Malondialdehyde (MDA) Content, and Total Antioxidant Capacity (T-AOC) Activity Assays

Semen from four FA treatments and one control group was used to analyze the ROS and MDA content and the T-AOC activity. 

The ROS content was measured using the reactive oxygen species assay kit (Nanjing Jiancheng Bioengineering Institute, Jiangsu, China), following the manufacturer’s protocol. In brief, 100 μL of semen was taken in a centrifuge tube and centrifuged at 1000× *g* for 10 min; the supernatant was discarded. Next, 500 μL of diluted 10 μmol/L DCFH-DA was added to resuspend the precipitate in a water bath at 37 °C for 30 min. At the end of the incubation period, the samples were centrifuged at 1000× *g* for 10 min, and the supernatant was discarded. The precipitate was washed twice with PBS, and the cell precipitate was collected via centrifugation; then, 600 μL PBS was added to resuspend the precipitate, and 20 μL suspension was added to the 96-well plate to measure the fluorescence value. A multifunctional enzyme standard was used; the excitation wavelength was set at 488 nm and the emission wavelength at 525 nm. There were three replicates per group.

The MDA content was measured using the MDA assay kit (Nanjing Jiancheng Bioengineering Institute, Jiangsu, China) according to the manufacturer’s instructions. In brief, 100 μL of semen was mixed with 1 mL of MDA assay working solution and incubated in a water bath at 95 °C for 40 min. At the end of the incubation period, the samples were centrifuged at 4000× *g* for 10 min. Additionally, 200 μL of supernatant was added to the 96-well plate to measure the absorbance value at 530 nm and normalize the protein content (nmol/mg protein) using the BCA assay kit (Thermo Scientific, Waltham, MA, USA). There were three replicates per group. 

The T-AOC activity was determined using the T-AOC assay kit (Nanjing Jiancheng Bioengineering Institute, Jiangsu, China) according to the manufacturer’s instructions. In brief, 200 μL of semen was centrifuged at 800× *g* for 10 min, and the supernatant was discarded. The sperm pellets were collected and repeated twice using precooled PBS; 200 μL of precooled lysate was added, and the sperm pellets were shattered using ultrasonication, operated on ice; the corresponding reagents were added sequentially according to the kit instructions. Finally, the absorbance value at 520 nm was measured and normalized to the protein content (U/mg protein) using the BCA assay kit (Thermo Scientific, Waltham, MA, USA). There were three replicates per group. 

### 2.6. Fertility Test

To evaluate the effect of FA on fertility, semen containing 0 μmol/L and 50 μmol/L FA was stored at 17 °C for 3 days and injected into synchronized estrus goats (aged 2–3 years) via artificial insemination; fresh semen was injected as the control. The dose of semen used was approximately 5 × 10^8^ sperm/mL. A total of 54 estrus goats were inseminated at random and kept in the same feeding conditions and management environment. Early pregnancy was detected via ultrasound 35–40 days after artificial insemination.

### 2.7. Statistical Analysis

All the results are presented as the mean ± SE (*n* = 3). One-way ANOVA was used for statistical comparison using SPSS 23.0 software. *p* ≤ 0.05 was considered statistically significant.

## 3. Results

To investigate whether FA affects semen quality and antioxidant capacity during storage at 17 °C, sperm total motility, sperm plasma membrane integrity, sperm acrosome integrity, and ROS, MDA, and T-AOC content were evaluated at different times and at different FA concentrations, respectively.

### 3.1. Effect of Different Concentrations of FA on Sperm Total Motility of Black-Headed Goats

The effects of different concentrations of FA on black-headed goat sperm total motility during liquid storage at 17 °C are shown in Table 1. On days 1 and 3, the sperm total motility was significantly improved in the 25 and 50 μmol/L FA concentration groups compared with the control group (0 μmol/L, *p* < 0.05); on day 2, the sperm total motility of the 25, 50, and 100 μmol/L groups was significantly higher than that of the control group (*p* < 0.05); on day 4, the sperm total motility in the 25, 50, and 100 μmol/L groups was significantly higher than that in the control group (*p* < 0.05); on day 5, the sperm total motility was highest in the 50 μmol/L group (*p* < 0.05), and the sperm total motility of the 200 μmol/L group was lower than that of the control group, but there was no significant difference. Generally, the sperm total motility was significantly improved in the 25 and 50 μmol/L FA concentration groups.

### 3.2. Effect of Different Concentrations of FA on Sperm Plasma Membrane Integrity of Black-Headed Goats

The effect of different concentrations of FA on black-headed goat sperm plasma membrane integrity during liquid storage at 17 °C is shown in Table 2. On day 1, the 50 μmol/L group was significantly higher than that of the 0 and 100 μmol/L groups (*p* < 0.05); the plasma membrane integrity of the 50 μmol/L FA concentration group was significantly higher than that of the other groups on days 3 and 5 (*p* < 0.05), and there was no significant difference between the other groups; on day 2, the 50 μmol/L group was significantly higher than that of the 0, 100, and 200 μmol/L groups (*p* < 0.05). Meanwhile, there was no significant difference between the 50 μmol/L and 25 μmol/L groups. The plasma membrane integrity of the 50 μmol/L group was significantly higher than that of the 25 and 200 μmol/L groups on day 4 of semen preservation (*p* < 0.05). These results suggested that adding 50 μmol/L FA to the semen diluent could significantly improve the integrity of the sperm plasma membrane.

### 3.3. Effects of Different Concentrations of FA on the Sperm Acrosome Integrity of Black-Headed Goats

The effect of different concentrations of FA on black-headed goat sperm acrosome integrity during liquid storage at 17 °C is shown in Table 3. Sperm acrosome integrity was significantly improved only in the 50 μmol/L FA concentrations group compared with the 0 μmol/L group on days 1, 2, and 5 (*p* < 0.05); on 3 day, the sperm acrosome integrity in the 25 and 50 μmol/L groups was significantly higher than in the control group (*p* < 0.05); on day 4, sperm acrosome integrity was significantly improved in the 25, 50, and 100 μmol/L groups compared with the 0 μmol/L group.

### 3.4. Effects of Different Concentrations of FA on Sperm ROS Content in Black-Headed Goats

The effect of different concentrations of FA on black-headed goat sperm ROS content during liquid storage at 17 °C is shown in Table 4. The ROS content was significantly reduced in the 25, 50, and 100 μmol/L FA concentration groups compared with the control group (*p* < 0.05) and reached its lowest in the 50 μmol/L group on the third and fifth days of preservation. There was no significant difference between the 200 μmol/L group and the control group.

### 3.5. Effects of Different Concentrations of FA on Sperm MDA Content in Black-Headed Goats

The effect of different concentrations of FA on black-headed goat sperm MDA content during liquid storage at 17 °C is shown in Table 5. The MDA content was significantly reduced in the 50 μmol/L group (*p* < 0.05), while there were no significant differences among the other groups on day 3; on day 5, the MDA content of the 25 and 50 μmol/L groups was significantly decreased, while that of the 200 μmol/L group was significantly increased (*p* < 0.05).

### 3.6. Effects of Different Concentrations of FA on Sperm T-AOC Levels in Black-Headed Goats

The effects of different concentrations of FA on the T-AOC level of black-headed goat sperm during liquid storage at 17 °C are shown in Table 6. The T-AOC level was the highest in the 50 μmol/L FA concentration group either on the third or fifth day of preservation (*p* < 0.05).

### 3.7. Effects of Semen Stored at 17 °C for 3 Days on the Conception Rate of Black-Headed Goats

The effects of different semen kept in 17 °C liquid storage for 3 days on the conception rate of black-headed goats are shown in Table 7. There was no significant difference between the fresh semen group and the 50 μmol/L FA group (*p* < 0.05), while the 0 μmol/L FA group showed a significant decrease. 

## 4. Discussion

Semen preservation during liquid storage has the advantages of being a simple operation, reducing damage to sperm, and having low investment costs. However, the surface of the sperm cell membrane contains large amounts of polyunsaturated fatty acids (PUFA), which are susceptible to free radicals such as ROS attacks, shortening the sperm’s survival time [16,17]. Generally, seminal plasma contains a large amount of enzymatic and nonenzymatic antioxidants [18], which act simultaneously to neutralize free radicals and prevent further oxidative reactions [12]. In the process of semen preservation in vitro, its own antioxidants are insufficient to maintain semen quality for long periods of time, so it is crucial to add effective exogenous antioxidants to the semen extender [19]. 

Ferulic acid (FA) is a natural phenolic phytochemical that exists in plants in free or conjugated forms and can also be covalently bound to plant cell wall polysaccharides. FA has low toxicity and extensive pharmacological activities, such as antioxidant, antiallergic, anticancer, antibacterial, and anti-inflammatory properties, and plays a regulatory role in cell signaling and gene expression [20]. Some studies have reported that FA could be used as an exogenous antioxidant for semen preservation. For instance, FA protected the mitochondrial, acrosome, and plasma membrane integrity of stallion sperm after 8 h of storage at 4 °C [21]. FA is beneficial to human sperm viability and motility in both fertile and infertile individuals [22]. Trans-FA could ameliorate the toxic effect of β-cyfluthrin via the reduction of peroxidative and nitrosative reactions during the cold preservation (4 °C) of rooster semen [13]. Pei et al. [14] found that FA significantly improved the quality of frozen and thawed boar sperm. In this study, to test whether FA added at different concentrations to the semen extender can improve the preservation quality of goat buck semen during liquid storage at 17 °C, motility parameters such as sperm total motility, functional integrity factors such as plasma membrane integrity and acrosome integrity rates, and oxidative status parameters such as ROS, MDA, and T-AOC were evaluated. Our results demonstrated that adding an appropriate concentration of FA significantly improved sperm total motility, plasma membrane integrity, and acrosome integrity while reducing the peroxidation rate (as evidenced by the levels of ROS, MDA, and T-AOC). These results indicated that FA could alleviate oxidative damage during the liquid storage of goat buck semen and improve the quality of semen preservation. Abnormal sperm morphology, DNA damage, and lipid peroxidation affect fertilization and early embryonic development [23,24]. To further evaluate the effect of FA on semen quality, the conception rate of estrus goats was tested. Our results showed that the conception rate of goats was not significantly affected by semen stored at 17 °C with 50 μmol/L FA for 3 days. 

ROS, a class of single electrons of oxygen, comprises superoxide anions (O^2−^), hydrogen peroxide (H_2_O_2_), and hydroxyl radicals (OH^−^) [25]. The content of ROS increases significantly with preservation time, resulting in decreased sperm motility and both changes in membrane permeability and acrosome structure [26], reducing the quality of semen. MDA is the final product of free radicals and lipid peroxidation [27] and has been used in various biochemical assays to monitor the degree of peroxidative damage sustained by sperm [28]. Both ROS and MDA negatively affect semen quality. T-AOC activity could reveal the total antioxidant capacity of semen [29]. In this study, we chose ROS, MDA, and T-AOC as redox parameters to evaluate semen’s oxidative status.

It was noteworthy that the change in sperm total motility, plasma membrane integrity, acrosome integrity, and T-AOC levels showed a trend of rising first and then falling with the increase in FA concentration, while the contents of ROS and MDA decreased first and then increased. On the one hand, the high concentration of FA may influence the osmotic pressure of the extender, affecting the permeability of the sperm membrane, destroying the sperm structure, and reducing sperm progressive motility [30,31]. On the other hand, the high concentration of FA may be toxic to and damage sperm, and it may cause excessive activation of antioxidant enzymes and mitochondria, which affects the physiological state of the sperm [32]. Additionally, high amounts of antioxidative substances disturb redox balances and act as a pro-oxidant, increasing proinflammatory mediators of free radicals, stimulating oxidative toxicity, and nitrosylation of proteins [33,34,35,36].

The protective ability of antioxidants depends on their concentration, and lipid peroxidation is induced when antioxidants and ROS production are imbalanced [37,38]. Some studies have shown that FA has a dose-dependent protective effect on semen: low doses increased cell survival, while high doses increased apoptosis. Shayan-Nasr et al. [13] reported that trans-FA at 10 and 25 mM doses significantly restored the motility and viability of rooster spermatozoa at 4 °C. When exceeding the optimal concentration, FA not only failed to provide adequate protection but also had adverse effects on sperm viability and metabolites of the diluent, inducing elevated MDA content and reduced T-AOC levels. Pei et al. [14] reported that supplementation of the freezing extender with 0.25 mM or 0.35 mM FA has a beneficial effect on frozen and thawed boar sperm. The sperm viability, plasma membrane integrity, and acrosome integrity of thawed spermatozoa reached a maximum at the optimal addition of FA, while these parameters remained stable or even showed a significant decrease as the concentration continued to increase [14]. Salimi et al. [15] showed that 5 mM and 10 mM trans-FA improved the forward progressive motility and curvilinear velocity of ram semen at 4 °C; samples treated with 25 mM trans-FA showed the lowest total motility, forward progressive motility, and viability. In this study, we found that the optimal amount of FA added during the storage of black-headed goat semen at 17 °C was 50 μmol/L, which differed from the results of previous studies. This effect may be the result of different extenders, different dilution ratios, different animal species, and different storage procedures.

## 5. Conclusions

The results of this study showed that when 50 μmol/L FA was added to the extender, the sperm total motility, plasma membrane integrity, acrosome integrity, and T-AOC were significantly higher than those of the control group throughout the preservation period. Comprehensive indicators of all aspects showed that the concentration of 50 μmol/L FA was the best for the preservation of black-headed goat bucks’ semen at 17 °C; it could effectively alleviate the oxidative stress damage caused by ROS, prolong the survival time of sperm, improve the quality of semen preservation, and do so without affecting the goat conception rate within 3 days. This study provides an important reference value for the liquid preservation of goat semen and helps promote the application of AI technology in goat breeding.

## Figures and Tables

**Table 1 animals-13-02469-t001:** Effect of FA on the sperm total motility of black-headed goats (%).

FA Concentrations	Time of Storage				
	1 d	2 d	3 d	4 d	5 d
0 μmol/L	88.26 ± 0.44 ^b^	84.73 ± 2.18 ^b^	83.27 ± 0.17 ^c^	77.97 ± 0.67 ^c^	77.13 ± 0.29 ^d^
25 μmol/L	92.33 ± 0.24 ^a^	87.40 ± 0.72 ^a^	86.13 ± 0.29 ^a^	83.60 ± 0.23 ^a^	80.63 ± 0.66 ^b^
50 μmol/L	92.90 ± 0.29 ^a^	87.70 ± 0.20 ^a^	85.10 ± 0.06 ^ab^	84.20 ± 0.17 ^a^	82.57 ± 0.29 ^a^
100 μmol/L	88.17 ± 0.65 ^b^	86.93 ± 0.44 ^a^	83.70 ± 0.95 ^bc^	81.40 ± 1.37 ^ab^	79.07 ± 0.42 ^c^
200 μmol/L	89.43 ± 1.57 ^b^	85.73 ± 0.58 ^ab^	83.30 ± 1.90 ^c^	80.23 ± 0.52 ^bc^	76.63 ± 0.35 ^d^

Note: For the same column data, different lowercase superscript letters indicate significant differences (*p* < 0.05), while the same letters indicate no significant differences (*p* > 0.05).

**Table 2 animals-13-02469-t002:** Effects of FA on the sperm plasma membrane integrity of black-headed goats (%).

FA Concentrations	Time of Storage				
	1 d	2 d	3 d	4 d	5 d
0 μmol/L	78.34 ± 1.07 ^b^	74.30 ± 0.86 ^bc^	72.27 ± 0.73 ^c^	70.50 ± 0.64 ^abc^	64.18 ± 1.15 ^bc^
25 μmol/L	80.37 ± 1.22 ^ab^	76.71 ± 1.29 ^ab^	73.52 ± 1.34 ^b^	69.44 ± 1.03 ^bc^	64.58 ± 0.52 ^bc^
50 μmol/L	81.50 ± 1.32 ^a^	78.93 ± 0.84 ^a^	78.10 ± 0.54 ^a^	72.79 ± 1.70 ^a^	71.49 ± 0.87 ^a^
100 μmol/L	78.33 ± 1.28 ^b^	75.29 ± 0.91 ^bc^	72.48 ± 1.14 ^bc^	71.21 ± 0.49 ^ab^	65.52 ± 0.59 ^b^
200 μmol/L	79.74 ± 0.70 ^ab^	73.61 ± 1.01 ^c^	71.67 ± 0.70 ^c^	68.48 ± 1.00 ^c^	62.30 ± 1.13 ^c^

Note: For the same column data, different lowercase superscript letters indicate significant differences (*p* < 0.05), while the same letters indicate no significant differences (*p* > 0.05).

**Table 3 animals-13-02469-t003:** Effects of FA on the sperm acrosome integrity of black-headed goats (%).

FA Concentrations	Time of Storage				
	1 d	2 d	3 d	4 d	5 d
0 μmol/L	92.11 ± 0.70 ^b^	88.65 ± 0.83 ^b^	86.59 ± 0.70 ^c^	83.66 ± 0.66 ^c^	81.39 ± 0.70 ^bc^
25 μmol/L	93.05 ± 0.54 ^ab^	90.51 ± 0.81 ^ab^	89.21 ± 0.64 ^ab^	86.37 ± 0.77 ^b^	83.44 ± 0.73 ^b^
50 μmol/L	94.22 ± 0.50 ^a^	91.34 ± 0.66 ^a^	89.62 ± 0.86 ^a^	88.39 ± 0.79 ^a^	86.34 ± 0.29 ^a^
100 μmol/L	92.03 ± 0.68 ^b^	89.32 ± 0.75 ^ab^	87.42 ± 0.45 ^bc^	85.63 ± 0.61 ^b^	82.32 ± 0.81 ^bc^
200 μmol/L	92.75 ± 0.52 ^ab^	88.84 ± 0.95 ^b^	87.27 ± 0.63 ^c^	85.03 ± 0.45 ^bc^	81.16 ± 0.54 ^c^

Note: For the same column data, different lowercase superscript letters indicate significant differences (*p* < 0.05), while the same letters indicate no significant differences (*p* > 0.05).

**Table 4 animals-13-02469-t004:** Effects of FA on the sperm ROS content of black-headed goats.

FA Concentrations	Time of Storage	
	3 d	5 d
0 μmol/L	2631.99 ± 13.81 ^a^	3092.3 ± 23.39 ^a^
25 μmol/L	2258.46 ± 57.56 ^b^	2763.69 ± 15.68 ^b^
50 μmol/L	2026.3 ± 42.47 ^c^	2044.18 ± 18.53 ^c^
100 μmol/L	2148.54 ± 12.47 ^bc^	2523.23 ± 22.8 ^b^
200 μmol/L	2643.61 ± 47.78 ^a^	3223.03 ± 36.11 ^a^

Note: For the same column data, different lowercase superscript letters indicate significant differences (*p* < 0.05), while the same letters indicate no significant differences (*p* > 0.05).

**Table 5 animals-13-02469-t005:** Effects of FA on sperm MDA content of black-headed goats (nmol/mg protein).

FA Concentrations	Time of Storage	
	3 d	5 d
0 μmol/L	2.29 ± 0.08 ^a^	2.68 ± 0.14 ^b^
25 μmol/L	2.10 ± 0.08 ^ab^	2.37 ± 0.13 ^c^
50 μmol/L	1.92 ± 0.10 ^b^	2.28 ± 0.08 ^c^
100 μmol/L	2.21 ± 0.05 ^a^	2.79 ± 0.09 ^b^
200 μmol/L	2.35 ± 0.12 ^a^	2.91 ± 0.09 ^a^

Note: For the same column data, different lowercase superscript letters indicate significant differences (*p* < 0.05), while the same letters indicate no significant differences (*p* > 0.05).

**Table 6 animals-13-02469-t006:** Effects of FA on the sperm T-AOC level of black-headed goats (U/mg protein).

FA Concentrations	Time of Storage	
	3 d	5 d
0 μmol/L	1.44 ± 0.01 ^b^	1.22 ± 0.04 ^ab^
25 μmol/L	1.63 ± 0.07 ^ab^	0.94 ± 0.04 ^c^
50 μmol/L	1.69 ± 0.08 ^a^	1.42 ± 0.02 ^a^
100 μmol/L	1.59 ± 0.04 ^b^	1.19 ± 0.03 ^abc^
200 μmol/L	1.45 ± 0.01 ^b^	1.03 ± 0.02 ^bc^

Note: For the same column data, different lowercase superscript letters indicate significant differences (*p* < 0.05), while the same letters indicate no significant differences (*p* > 0.05).

**Table 7 animals-13-02469-t007:** Effect of different semen on the conception rate of black-headed goats.

Group	Inseminated Goats (n)	Pregnant Goats (n)	Conception Rate (%)
Fresh semen	18	15	83.33 ^a^
0 μmol/L FA	16	7	43.75 ^b^
50 μmol/L FA	20	16	80.00 ^a^

Note: Different lowercase letters among groups indicate significant differences (*p* < 0.05), while the same letters indicate no significant differences (*p* > 0.05).

## Data Availability

The data presented in this study are available upon request from the corresponding author.

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
