# Peer review of "Effect of Ferulic Acid on Semen Quality of Goat Bucks during Liquid Storage at 17 °C"

_animals, 2023, doi:10.3390/ani13152469_

Round 1
Reviewer 1 Report
The manuscript “Effect of ferulic acid on semen quality of goat during normal
temperature storage” has scientific merit and is interesting, but it is poorly presented, so it needs to be restructured and submitted again.
The manuscript needs a thorough review of written English, as there are many grammar mistakes. There is an incorrect usage of articles “a”, “an”, and “the”; there is also a mixup in the plural and singular uses.
Besides, the materials and methods section needs some clarifications; it does not specify if only one ejaculate of each buck was used or more. Also, information regarding the does used is necessary; it doesn’t specify the number of animals or their characteristics; please elaborate on this.
For the results section, don’t mix the discussion, introduction, or materials and methods information; only show the results obtained (i.e., L167, L190-193, L195-197).
The main issue of the manuscript is regarding the discussion. Most of it seems like a literature review when it should discuss the finding and make an attempt to explain the results, the reasons why everything that happened happened, and support it with other research. Something like the paragraph found in L242-253.
The manuscript needs a thorough review of written English, as there are many grammar mistakes. There is an incorrect usage of articles “a”, “an”, and “the”; there is also a mixup in the plural and singular uses.
Author Response
Dear Reviewer,
Thanks for your comments concerning our manuscript entitled ‘Effect of ferulic acid on semen quality of goat during normal temperature storage’. The comments were all valuable and very helpful for revising and improving our manuscript, as well as for providing important guidance regarding the significance of our studies. We have carefully reviewed the comments and have made corrections that we hope will meet with approval. The revised portions of the manuscript are marked in red.
Point 1: The manuscript needs a thorough review of written English, as there are many grammar mistakes. There is an incorrect usage of articles “a”, “an”, and “the”; there is also a mixup in the plural and singular uses.
Response 1: Thanks for your comment. Our manuscript has undergone English language editing by MDPI (https://www.mdpi.com/authors/english).
Point 2: Besides, the materials and methods section needs some clarifications; it does not specify if only one ejaculate of each buck was used or more. Also, information regarding the does used is necessary; it doesn’t specify the number of animals or their characteristics; please elaborate on this.
Response 2: Thanks for your comment. These were added in the Materials and Methods section: the semen of three black-headed goat bucks (3–5 years old) was used in this study. Ejaculates were collected from the bucks using an artificial vagina in the presence of estrus goats, two times per week during the breeding seasons. The black-headed goat is a new breed based on Macheng Black Goat (Chinese local breed) and introduces Boer Goat line-age. It has a black head and a white body. One ejaculate of each goat buck was collected. The semen volume of each goat buck collected was from 0.5~1.5 mL every time. The semen used was milky white and had no abnormal smell. The sperm concentration was >2×109 sperm/mL and the motility >0.8. The semen was pooled to minimize individual differ-ences between goat bucks (Lines 75-83).
Point 3: For the results section, don’t mix the discussion, introduction, or materials and methods information; only show the results obtained (i.e., L167, L190-193, L195-197).
Response 3: Thanks for your comment. The discussion, introduction, or materials and methods information were deleted in the results section (Lines 200, 229-232)
Point 4: The main issue of the manuscript is regarding the discussion. Most of it seems like a literature review when it should discuss the finding and make an attempt to explain the results, the reasons why everything that happened happened, and support it with other research. Something like the paragraph found in L242-253.
Response 4: Thanks for your comment. We have made significant modifications to the discussion section.

Reviewer 2 Report
Effect of ferulic acid on the quality of goat semen stored at normal temperature was investigated in this study. Improving semen quality of goat is an interesting topic and challenging work, both cryopreservation and normal temperature preservation. There are some issues needed to be improved before this manuscript published. 1. English expression needs much improvement. 2. normal temperature=17℃=room temperature? Why 17℃ condition was selected to preserve the semen? 3. The data was showed as "means±SE" but not "means±SEM". Why the effect of Time of storage was not analyzed? Two-way ANOVA should be used for statistical comparisons. 4. Discussion should be the analysis of the results by referring to basic theory and previous related results but not pure literature review. 5. Why semen stored at normal temperature for 3 days was selected in artificial insemination trial but not others?
Many grammars should be improved, e.g., the missing of verbs, the misuse of conjunction.
Author Response
Dear Reviewer,
Thanks for your comments concerning our manuscript entitled ‘Effect of ferulic acid on semen quality of goat during normal temperature storage’. The comments were all valuable and very helpful for revising and improving our manuscript, as well as for providing important guidance regarding the significance of our studies. We have carefully reviewed the comments and have made corrections that we hope will meet with approval. The revised portions of the manuscript are marked in red.
Point 1: English expression needs much improvement.
Response 1: Thanks for your comment. Our manuscript has undergone English language editing by MDPI (https://www.mdpi.com/authors/english).
Point 2: normal temperature=17℃=room temperature? Why 17℃ condition was selected to preserve the semen?
Response 2: Thanks for your comment and questions. Perhaps room temperature is more appropriate, for simplicity and clarity, 17℃ was directly used instead of room temperature or normal temperature throughout the full text. Semen preservation in goats is less studied and the technology is still not mature, the optimum temperature for liquid storage of goat semen is uncertain, referring to the optimal temperature of boar semen is 17℃, we chose 17℃ in this study.
Point 3: The data was showed as "means±SE" but not "means±SEM". Why the effect of Time of storage was not analyzed? Two-way ANOVA should be used for statistical comparisons.
Response 3: Thanks for your comment and question. We have changed the "means±SEM" to the "means±SE". In this study, we focused on the effect of different concentrations of ferulic acid (FA) on the quality of goat semen at the same time point, so the effect of Time of storage was not analyzed. And, One-way ANOVA was used for statistical comparison, which was widely used in similar studies, such as:
[1] Zhang, L.M.; Wang, Y.H.; Sohail, T.; Kang, Y.; Niu, H.Y.; Sun, X.M.; Ji, D.J.; Li, Y.J. Effects of Taurine on Sperm Quality during Room Temperature Storage in Hu Sheep. Animals. 2021, 11, 2725.
[2] Zhang, X.G.; Liu, Q.; Wang L.Q.; Yang G.S.; Hu, J.H. Effects of glutathione on sperm quality during liquid storage in boars. Anim Sci J. 2016, 87, 1195-1201.
Point 4: Discussion should be the analysis of the results by referring to basic theory and previous related results but not pure literature review.
Response 4: Thanks for your comment. We have made significant modifications to the discussion section.
Point 5: Why semen stored at normal temperature for 3 days was selected in artificial insemination trial but not others?
Response 5: Thanks for your comment and question. It is best to test fertility at each storage time point, but in the actual breeding process, we carry out artificial insemination in batches, usually every 3 days. Therefore, we have chosen 3 days. This may be our innovation point, because most of the literature on semen preservation only detects some physicochemical parameters, the fertility test was not performed.

Reviewer 3 Report
Comments in attachment.

English language must be reviewed.
Author Response
Dear Reviewer,
Thanks for your comments concerning our manuscript entitled ‘Effect of ferulic acid on semen quality of goat during normal temperature storage’. The comments were all valuable and very helpful for revising and improving our manuscript, as well as for providing important guidance regarding the significance of our studies. We have carefully reviewed the comments and have made corrections that we hope will meet with approval. The revised portions of the manuscript are marked in red.
Point-by-point responses to the reviewer’ comments:
General comment:
The manuscript reports the results on the effect of adding an antioxidant compound, ferulic acid, to the dilution and preservation medium of goat bucks semen, stored at 17ºC for 5 days. The subject of the study is interesting, given the need to be able to preserve and/or transport good quality breeder semen for use in AI. The study is simple and the manuscript is easy to read and follow. However, it lacks key information in all the chapters that make it difficult to understand its objective. Nor is there a hypothesis that favors the understanding of the meaning of the study and its adequate justification. The mere absence of studies on a particular specie is, in this case, a weak argument.
Response: Thanks for your comment. We have reorganized the entire text and added missing key information in all the chapters. To explain the meaning of the study and its adequate justification, some information was added in Introduction (Lines 37-42).
Specific comments:
Title
The title does not adequately represent the study, as there is no conventional definition of "normal temperature". I think it would look much better as follows:
"Effect of ferulic acid on semen quality of goat bucks during storage at 17ºC"
Response: Thanks for your comment and suggestion. The title has been changed to "Effect of ferulic acid on semen quality of goat bucks during liquid storage at 17ºC".
Introduction
This chapter presents general information on the subject, but does not explain the need to preserve/improve the quality of goat buck semen stored at 17ºC, which is a key aspect to assess the scientific-technical potential and the usefulness of the results.
- Lines 38-39: I think there is a conceptual mistake in the sentence, since normally the literature on semen conservation refers to cryopreservation when the semen is subjected to different freezing temperatures and refrigeration when the semen is kept at temperatures above 0ºC, but generally below 15ºC. Additionally, there is literature on semen storage at room temperature, which is generally considered to be between 23-25ºC, or else the specific temperature at which the study was conducted is identified. As explained previously, the term "normal temperature" is vague and needs to be reconsidered along the text.
Response: Thanks for your comment. It seems that the boundaries of temperature are not consistent, 15 ºC was also defined as room temperature in some literature[1]. Some literature suggests it can be divided into liquid and frozen preservation, according to the preservation temperature[2,3]. Liquid preservation has the advantages of simple operation, low requirements for preservation conditions, and a good effect on fertility compared with frozen preservation (Lines 45-49). So, we adopted this view in this study. Additionally, Semen preservation in goats is less studied and the technology is still not mature, the optimum temperature for liquid storage of goat semen is uncertain, referring to the optimal temperature of boar semen is 17℃, we chose 17℃ in this study.
[1]Zhang, L.M.; Wang, Y.H.; Sohail, T.; Kang, Y.; Niu, H.Y.; Sun, X.M.; Ji, D.J.; Li, Y.J. Effects of Taurine on Sperm Quality during Room Temperature Storage in Hu Sheep. Animals. 2021, 11, 2725.
[2] Salamon, S.; Maxwell, W.M. Storage of ram semen. Anim Reprod Sci. 2000, 62, 77-111.
[3]Shayan-Nasr, M.; Ghaniei, A.; Eslami, M.; Zadeh-Hashem, E. Ameliorative role of trans-ferulic acid on induced oxidative toxicity of rooster semen by β-cyfluthrin during low temperature liquid storage. Poult Sci. 2021, 100, 101308.
-Line 54: the “in vitro” concept and any other of Greco-Roman roots must be written in italics.
Response: Thanks for your comment. “in vitro” has been written in italics (Line 61).
- Lines 58-59: The sentence is imprecise, since only the study of reference 14 was carried out with frozen semen, while those of references 13 and 15 were carried out in refrigeration at 4 ºC, then "…a positive role in the cryopreservation…" cannot be mentioned.
Response: Thanks for your comment. “cryopreservation” has been deleted in line 66.
At the end of this chapter, at least one objective should be clearly mentioned.
Response: Thanks for your comment. The objective was presented step by step: first, with the development of science and technology, artificial insemination (AI) is seeing wider use in goat breeding. The success of AI is largely dependent on the quality of preserved semen, which is a key factor affecting the effectiveness of AI. However, compared with other livestock, semen preservation in goats is less studied and the technology is still not mature. Thus, we are seeking methods that can be applied to the preservation of goat buck semen (Lines 37-42); then, to test whether FA can improve the preservation quality of goat buck semen during liquid storage, this study was carried out (Lines 68-72).
Materials and Methods
- Line 68 and along the manuscript: Please, replace the term “rams” (used for male sheep) for “bucks” or “goat bucks” (male goat).
Response: Thanks for your comment. “rams” has been replaced in the whole text.
- Line 73: Delete the word “The” at beginning of sentence.
Response: Thanks for your comment. “The” has been deleted in line 84.
- Lines 76-78: Please, justify briefly the selected doses of FA.
Response: Thanks for your comment. The selected doses of FA has been briefly expressed in lines 87-89.
- Line 79: Why was this temperature chosen? It is curious, because it is not a classic refrigeration temperature, nor is it a frequent room temperature. Besides, the Introduction does not justify the choice of this temperature.
Response: Thanks for your comment. Semen preservation in goats is less studied and the technology is still not mature, the optimum temperature for liquid storage of goat semen is uncertain, referring to the optimal temperature of boar semen is 17℃, we chose 17℃ in this study.
- Line 82: Why was the sample taken randomly? please explain. How many tubes or replicates were obtained for each treatment?
Response : Thanks for your comment and questions. “randomly” indicating that the selected samples were well mixed and not targeted, so it is more representative. But to avoid ambiguity, “randomly” has been deleted in line 92. There were three replicates per group (lines 98, 120, 136, 140).
- Line 85: Please, fully identify the equipment (model, manufacturer, city and country of manufacture).
Response : Thanks for your comment. These has been added in lines 95-96.
- Line 86 and along the entire manuscript: replace the term “viability” for “motility”. As described, only sperm motility was evaluated, not sperm viability.
Response : Thanks for your comment. “viability” has been replaced in the whole text.
- Lines 92-93: “Three or more fields of view were randomly selected…”; How many more?
Response: Thanks for your comment and question. “How many more” were uncertain, as the number of sperm in each group was different, but the total number of sperm observed was at least 200.
- Lines 93-94: “…total number of sperm observed was at least 200.”; By sample? How many replicates per group?. Please, explain clearly.
Response: Thanks for your comment and question. Four FA treatments and one control group (Line 101). There were three replicates per group (Line 106).
- Lines 101: Please, in the centrifugation indicate the force used, not the rpm.
Response: Thanks for your comment. Centrifugation at 1,500 x g (Line 114).
- Line 106: “Three or more fields…”; How many more?.
Response: Thanks for your comment and question. “How many more” were uncertain, as the number of sperm in each group was different, but the total number of sperm observed was at least 200.
- Lines 106-107: “…the total number of sperm observed was at least 200…”; per replica? per sample? per group?, please clarify.
Response: Thanks for your comment and question. 10 μL semen from four FA treatments and one control group was used to analyze the sperm acrosome integrity (Lines 108-109). There were three replicates per group (Line 120-121).
- Line 110: What kind of sample and how many replicates were used? How many duplicates per sample were used?
Response: Thanks for your comment and questions. Experiments from 2.2 to 2.6 were all based on 2.1(Semen collection and liquid preservation): the semen of three black-headed goat bucks (3–5 years old) was used in this study (Line 75); The semen was pooled to minimize individual differences between goat bucks (Lines 82-83); The mixed semen was diluted 10-fold with diluent and divided equally into five aliquots. FA was added to the base extender at concentrations of 25, 50, 100, and 200 μmol/L, while the control was the base extender without FA. All samples were stored in a constant temperature refrigerator at 17°C (shaken and turned over every 12 h) and used for the experiment (Lines 86-90). Semen from four FA treatments and one control group was used to analyze the ROS and MDA content and the T-AOC activity (Lines 124-125). There were three replicates per group (Lines 135-136).
- Line 117: Why was only one treatment used in this trial, which is perhaps the most relevant from a reproductive point of view?.
- Line 118: Why was the treatment not tested for 5 days? This is perhaps the most relevant data to know for how many days the FA-treated semen could be stored to maintain adequate fertility. In addition, please replace the term “ewes” for “goats” in this line and along the text. How many goats per treatment? What was the dose of semen used? Were its natural heat or synchronized? Age of the goats?
Response: Thanks for your comment and questions. It is best to test fertility at each storage time point, but in the actual breeding process, we carry out artificial insemination in batches, usually every 3 days. Therefore, we have chosen 3 days. This may be our innovation point, because most of the literature on semen preservation only detects some physicochemical parameters, the fertility test was not performed; “ewes” has been replaced in the whole text; The dose of semen used was approximately 5×108 sperm/mL; A total of 54 estrus goats were inseminated; synchronized estrus goats (aged 2-3 years)(Lines 144-146).
- Line 124: “…P value of < 0.05…” or ≤ 0.05.
Response: Thanks for your comment. P≤ 0.05 was considered statistically significant (Lines 151-152).
Results
- Lines 131-132 and along the entire manuscript: Sperm viability was not evaluated, then replace for total motility.
Response: Thanks for your comment. “viability” has been replaced in the whole text.
- Line 143, Table 1 and 157 Table 2: “… longitudinal data…”This is not understood, perhaps it is better replace it for "column data”.
Response: Thanks for your comment. "column data” has been used in Lines 170, 186, 197, 207,…
- Line 166: In sperm? semen? please clarify this title.
- Line 172: where? Sperms? semen? please clarify the table title.
- Line 173: In sperm? semen? please clarify this title.
- Line 181: In sperm? semen? please clarify this title.
Response: Thanks for your comment and questions. All in sperm (Lines 199, 206, 209, 216, 219, 225).
- Line 188: Please, reconsider “normal temperature” in the this title.
Response: Thanks for your comment. “normal temperature” has been changed to 17°C (Line 228).
- Table 7, column headers: “Hibridization number” and “ Conception number” do not represent what you want to show. I believe the headings "Inseminated goats" and "pregnant goats" are more explanatory, respectively.
Response: Thanks for your comment. "Inseminated goats" and "pregnant goats" were used in Table 7.
Discussion
In this chapter, the first three paragraphs are more of a literature review than a discussion of the results, so the discussion is really very brief. Therefore, I suggest rewriting this chapter projecting the results based on the objective and the information present in the literature.
Response: Thanks for your comment and suggestion. We have made significant modifications to the discussion section.
- Line 245: But the addition of any solute could change the osmotic pressure… Could you also think of a reductive stress?
- Lines 250-251: This contradicts the previous sentence.
Response: Thanks for your comment and suggestion. The reasons may be: On the one hand, the high concentration of FA may influence the osmotic pressure of the extender, affecting the permeability of the sperm membrane, destroying the sperm structure, and reducing sperm progressive motility. On the other hand, the high concentration of FA may be toxic to and damage sperm, and it caused the excessive activation of antioxidant enzymes and mitochondria, which affected the physiological state of the sperm. Additionally, high amounts of antioxidative substances disturb redox balances and act as a pro‐oxidant, increasing pro‐inflammatory mediators of free radicals, stimulation of oxidative toxicity, and nitrosylation of proteins (Lines 282-290).
- Line 260: The FA concentrations cited in ref. 14 are several orders of magnitude higher than those used in the present experiment...please discuss this.
Response: Thanks for your comment. This effect may be the result of different extenders, different dilution ratios, different animals, and different storage procedures (Lines 308-310).
Conclusion
The conclusion summarizes some aspects of the results, but there is no projection of them. In practice, is it really advisable to use FA to preserve goat semen?
Response: Thanks for your comment. We have rewrited this chapter, and the results of this study showed that FA coule be used to preserve goat semen.
Additionally, the manuscript requires a thorough review of syntax and punctuation.
Response : Thanks for your comment. Our manuscript has undergone English language editing by MDPI (https://www.mdpi.com/authors/english).

Round 2
Reviewer 2 Report
If the effect of storage duration was analyzed, this study would give a specific suggestion on how long the liquid storage of goat buck semen lasted without negative effect. In a whole, the manuscript has been much improved and can be published after careful check.
Much better after revision.
Author Response
Dear Reviewer,
Thank you very much for your recognition of our research. We have carefully reviewed the comments (Round 2) and explained it below, we hope will meet with approval. Some other revised portions of the manuscript were highlighted in yellow.
Point 1: If the effect of storage duration was analyzed, this study would give a specific suggestion on how long the liquid storage of goat buck semen lasted without negative effect.
Response 1: Thanks for your comment. In this study, our results showed that supplementing semen with 50 μmol/L FA preserved at 17°C for 3 days had no significant effect on fertility by detecting the conception rate of black-headed goat, which were shown in Simple Summary (lines 18-19), Abstract (line 31), Discussion (lines 282-283), and Conclusions (lines 332-333).

Reviewer 3 Report
Please, consider the comments and suggestions noted on the attached version.

English language can be improved in some paragraphs.
Author Response
Dear Reviewer,
Thank you very much for your recognition of our research. We have carefully reviewed the comments (Round 2) and made corrections that we hope will meet with approval. The revised portions of the manuscript were highlighted in yellow.
Point 1: This is not clear, since lines 76-77 mention that ejaculates were obtained twice a week. Please clarify.
Response 1: Thanks for your comment. This was rewrited in lines 76-80.
Point 2:¿total motility?. According to what was described in the methodology, what was evaluated was total motility. This should be clearly stated in all chapters and tables of the manuscript.
Response 2: Thanks for your comment. “motility “has been replaced in the whole text.
Response 3: Thanks for your comment. “μL” was used in line 133.
Point 4: Were these evaluations also made using the sperm pellet? I think a brief description of the methodology could be made for clarity.
Response 4: Thanks for your comment and question. The methodology was added in lines 137-153.
Point 5: City, country?
Response 5: Thanks for your comment and question. These were added in lines 138 and 146.
Response 6: “n mol/mg” was changed into “nmol/mg” (line 143).
Response 7: “detected” was changed into “evaluated” (line 168).
Response 8: “different lowercase superscript” were used in lines 183, 200, 211, 221, 231, and 239.
Response 9: The results of 3.2 and 3.3 were reviewed, the revised portions were highlighted in yellow.
Response 10: “the antioxidant effects of the other groups were not significant compared to the
control group” was deleted an the end of line 237.
Response 11: “among groups” was added in line 247.
Response 12: “in vitro” in italics in line 256.
Response 13: “physiological” was changed into “pharmacologycal” in line 261.
Response 14: “added at different concentrations to the semen extender”, “storage at 17°C” were added in lines 270-272. There are also some changes in lines 273-276 in accordance with reviewer advice.
Response 15: In this study, we chose ROS, MDA, and T-AOC as the oxidative stress parameters to evaluate semen quality” was changed into “In this study, we chose ROS, MDA, and T-AOC as redox parameters to evaluate semen oxidative status” in line 291-292.
Response 16: “Notably, the sperm motility, plasma membrane integrity, acrosome integrity, and T-AOC levels increased and then decreased with the increase in FA concentration, while the ROS and MDA content decreased first and then increased” was changed into “It was noteworthy that the change of the sperm total motility, plasma membrane integrity, acrosome integrity, and T-AOC levels showed a trend of rising first and then falling with the increase of FA concentration, while the contents of ROS and MDA decreased first and then increased‘’in lines 293-296.
Response 17: “and it caused the excessive activation of antioxidant enzymes and mitochondria, which affected the physiological state of the sperm “ was changed into “and it may cause the excessive activation of antioxidant enzymes and mitochondria, which affected the physiological state of the sperm”.
Response 18: “animals”was changed into “animal species” in line 323.
Response 19: “ In conclusion” was deleted in line 326.
